# Personalized Medicine in Infant Population with Cancer: Pharmacogenetic Pilot Study of Polymorphisms Related to Toxicity and Response to Chemotherapy

**DOI:** 10.3390/cancers15051424

**Published:** 2023-02-23

**Authors:** Andrea Urtasun, Gladys G. Olivera, Luis Sendra, Salvador F. Aliño, Pablo Berlanga, Pablo Gargallo, David Hervás, Julia Balaguer, Antonio Juan-Ribelles, María del Mar Andrés, Adela Cañete, María José Herrero

**Affiliations:** 1Pediatrics Department, Pediatric Oncology Unit, University Clinic of Navarra, Av. de Pío XII, 36, 31008 Pamplona, Spain; 2Pediatric Oncology Unit, Hospital Universitario y Politécnico La Fe, Av. Fernando Abril Martorell 106, 46026 Valencia, Spain; 3Department of Pharmacology, Faculty of Medicine, Universitat de València, Av. Blasco Ibáñez 15, 46010 Valencia, Spain; 4Pharmacogenetics and Gene Therapy Platform, IIS La Fe, Torre A-Lab 4.03, Av. Fernando Abril Martorell 106, 46026 Valencia, Spain; 5Department of Pediatric and Adolescent Oncology, Institute Gustave Roussy Center, Rue Edouard Vaillant 114, 94800 Villejuif, France; 6Health in Code Group, Oncology Department, 46980 Paterna, Spain; 7Department of Applied Statistics and Operations Research and Quality, Universitat Politècnica de València, Camino de Vera, s/n, 46022 Valencia, Spain

**Keywords:** pharmacogenetics, SNP (single nucleotide polymorphism), chemotherapy, infant, toxicity, therapeutic efficacy, overall survival, event-free survival, anemia, neutropenia, thrombocytopenia

## Abstract

**Simple Summary:**

Cancer is still the leading cause of disease-related death in pediatric populations and an important source of morbidity. These can be due to the neoplasm itself but also to the treatment administered; infants are patients of special vulnerability. The aim of this work was to identify genetic variants that correlate with significant impacts on treatments’ safety and efficacy. 64 oncologic infant patients (under 18 months of age) were genotyped, calling for SNPs (single nucleotide polymorphisms) in genes related to the efficacy and/or toxicity of the chemotherapeutic drugs employed. The relationship of these genetic variants with 37 clinical parameters during 578 chemotherapy cycles was analyzed. Associations were found between 46 SNPs (in genes involved in drug transport and metabolism, gene repair, tumor suppression, and other biological functions) and survival and hematological toxicity. Following studies that confirm these findings, personalized medicine could be offered to improve response and tolerance to chemotherapeutic schemes.

**Abstract:**

Background: Pharmacogenetics is a personalized medicine tool that aims to optimize treatments by adapting them to each individual’s genetics, maximizing their efficacy while minimizing their toxicity. Infants with cancer are especially vulnerable, and their co-morbidities have vital repercussions. The study of their pharmacogenetics is new in this clinical field. Methods: A unicentric, ambispective study of a cohort of infants receiving chemotherapy (from January 2007 to August 2019). The genotypes of 64 patients under 18 months of age were correlated with severe drug toxicities and survival. A pharmacogenetics panel was configured based on PharmGKB, drug labels, and international experts’ consortiums. Results: Associations between SNPs and hematological toxicity were found. Most meaningful were: *MTHFR* rs1801131 GT increasing the anemia risk (OR 1.73); rs1517114 GC, *XPC* rs2228001 GT, increasing neutropenia risk (OR 1.50 and 4.63); *ABCB1* rs1045642 AG, *TNFRSF11B* rs2073618 GG, *CYP2B6* rs4802101 TC and *SOD2* rs4880 GG increasing thrombocytopenia risk (OR 1.70, 1.77, 1.70, 1.73, respectively). Regarding survival, *MTHFR* rs1801133 GG, *TNFRSF11B* rs2073618 GG, *XPC* rs2228001 GT, *CYP3A4* rs2740574 CT, *CDA* rs3215400 del.del, and *SLC01B1* rs4149015 GA were associated with lower overall survival probabilities (HR 3.12, 1.84, 1.68, 2.92, 1.90, and 3.96, respectively). Lastly, for event-free survival, *SLC19A1* rs1051266 TT and *CDA* rs3215400 del.del increased the relapse probability (HR 1.61 and 2.19, respectively). Conclusions: This pharmacogenetic study is a pioneer in dealing with infants under 18 months of age. Further studies are needed to confirm the utility of the findings in this work to be used as predictive genetic biomarkers of toxicity and therapeutic efficacy in the infant population. If confirmed, their use in therapeutic decisions could improve the quality of life and prognosis of these patients.

## 1. Introduction

The definition of "infant" in oncology is not homogeneous and varies depending on the collaborative group and the criteria employed (pharmacokinetic characteristics, tumor biology, therapeutic approaches, etc.). Although cancer in children older than 18 months can be considered a rare disease (or a group of rare diseases) due to its relatively low population incidence, cancer in the first stage of life is still rare. Oncological diseases in children under 12 months of age account for 11.4% of the total (about 100 cases per year in Spain), and only 15% of pediatric cancer is diagnosed in the first 18 months of life [1].

Global efficacy of pharmacologic treatments ranges from 25 to 60%, with chemotherapy achieving the lowest result [2]. Regarding toxicity, drug side effects represent the fourth cause of mortality in the USA and the sixth in the EU [3], besides being responsible for 30% of hospitalizations of patients over 55 years and provoking higher expenses than chronic conditions such as *diabetes mellitus* or cardiovascular diseases [4]. Specifically, in pediatric patients, the incidence of drug-related toxicity in hospitalized patients reaches 9.5% [5], with an average prevalence of 24% [6]. This is especially relevant in infants due to the severity of their side effects that could compromise their lives and the lack of studies involving the drug response (particularly with chemotherapeutic agents) in this population. In this work, we evaluate for the first time the role of individual genetic variability in under-18-month-old infants’ responses to chemotherapy drugs.

The medical findings of the adult population cannot be directly extrapolated to pediatrics due to the latter’s singularity at different levels. Infants present pharmacokinetic and pharmacodynamic differential features affecting drug management that must be considered and studied in detail: absorption differences (gastric pH variations, digestive emptying, absorption surface, quality of gut flora, muscle flow and its proportion, corneal and epidermal thickness); distribution differences (body composition, with more water and less fatty and muscular tissue that alters the management of hydrophobic/hydrophilic substances and the fraction of free drug, less restrictive hematoencephalic barrier that predisposes to neurotoxicity); differences in drug metabolism (variable maturity of important enzymatic complexes as CYP450); differences in excretion (lower kidney mass/blood flow, lower glomerular filtration, and alterations in tubule secretion) [7,8,9,10,11,12,13,14,15]. So, it becomes greatly important to perform pharmacogenetic studies in children, and particularly in infants, as the rules of the game change and we need to understand the different genes and SNPs that will be of relevance here differently than in adults.

In order to understand their relationship and maximize treatment efficacy while minimizing toxicity and long-term sequelae in this population, we evaluated potential associations between single nucleotide polymorphisms located in pharmacogenes and response to chemotherapeutic drugs in a cohort of 64 infants under 18 months of age.

## 2. Materials and Methods

### 2.1. Study Design

Ambispective observational study of correlation between genetic variants and clinical parameters in patients up to 18 months-old, diagnosed with oncological disease and having received or receiving chemotherapy at *Hospital Universitario y Politécnico La Fe* (Valencia, Spain). Retrospective recruitment comprises biobanked samples from January 2007 to September 2016, and prospective recruitment from October 2016 to August 2019. Clinical and analytical data (neoplasm, chemotherapy and other drugs, hematological support, clinical and analytical toxicity, epidemiological and diagnostic data, survival) from all the genotyped patients were available in the electronic or paper medical records (Figure 1). Data were abstracted from patients’ medical records by two independent pediatric oncologists.

Before genetic analysis, informed consent was obtained for each patient. Ethical approval was obtained from the *Comité de Ética de Investigación Clínica del Hospital Universitario y Politécnico La Fe de Valencia*, on 7 September 2016.

### 2.2. Clinical Parameters

#### 2.2.1. Drugs and Therapeutic Schemes

Protocols employed, chemotherapy administered in each cycle, and their doses are cited. Both the absolute and relative to body surface or weight doses were annotated (recommended based on age and protocol). Doses from each cycle were registered (Appendix A). If there was a difference between theoretical (according to protocol) and actually administered doses, this was marked in blue (Appendix A) if it was due to treatment toxicity in previous cycles.

Those patients who received transfusion support, omeprazole, and azole antifungals (possible enzymatic inducers/inhibitors) were annotated for the statistical analyses.

#### 2.2.2. Clinical and Analytical Toxicity. Survival Data

Antineoplastic drug-derived toxicities were grouped as qualitative variables (grades 0–5). The grade of toxicity assigned to each group was the maximum degree observed for the patient in all the specific toxicity categories within the group following the CTCAE 4.0 (Common Terminology Criteria for Adverse Events 4.0) classification scale. We evaluated the hematologic, infectious, digestive, renal, neurologic, respiratory, allergic, dermatologic, otologic, and ophthalmologic organic toxicities during the period of time corresponding to induction in solid tumors and previous to maintenance therapy or hematopoietic progenitor transplantation in the leukemia patients.

For survival data, overall survival and event-free survival were recorded from diagnosis to the last registered visit or *exitus*, ranging from 43 to 6.429 days. Data were censored to correctly perform the statistical analyses.

### 2.3. Genetic Variants

#### 2.3.1. SNPs Selection

The pharmacogenetics panel configuration (Appendix A) was based on PharmGKB database information, databases of scientific articles on this topic published from 1990 to the present (PubMed, MEDLINE, Cochrane Central Register), and journals in the pediatrics, oncology, and hematology fields.

#### 2.3.2. Samples and DNA Extraction

Sample collection: 2–5 mL of peripheral blood per patient was obtained from the Pediatric Oncohematology Department Biobank (this belongs to the national Biobanks Network of the Instituto de Salud Carlos III). It has the required license, reference PT13/0010/0026) or by direct extraction from the patient after informed consent obtention. It was kept frozen (at −20 °C) in EDTA solution.

DNA extraction: The “Nucleospin Blood” (Macherey Nagel, Düren, Germany) mini kit was used following the manufacturer’s instructions. DNA was eluted in TE (Tris-EDTA) and preserved upon its genotyping.

DNA quality control: the final concentration and purity of DNA were evaluated by spectrophotometry with a NanoDrop 2000 (NanoDrop Technologies Inc., Wilmington, DE, USA).

#### 2.3.3. Genotyping

Panel for genotyping was designed to include 62 polymorphisms that presented potential implication in patients’ treatment efficacy or drugs toxicity based on the bibliographic search performed (design named *VIP-Onco*). We could obtain suitable samples from 64 patients, and their genotyping was carried out at *Centro Nacional de Genotipado* (CEGEN, Santiago de Compostela, Spain) by mass spectrometry, employing a MassArray Analyzer (Agena Bioscience, San Diego, CA, USA).

### 2.4. Statistics

The statistical analyses were performed in collaboration with the Biostatistics and Data Science Unit at the Instituto de Investigación Sanitaria La Fe (Valencia, Spain). The correlation of polymorphisms with severe toxicity or overall survival (OS) and event-free survival (EFS) was calculated with logistic regression and Cox regression, respectively, both penalized with Elastic Net.

The association of clinical outcomes with pharmacogene variants was evaluated with R software (version 4.1.0) by fitting logistic or Cox regression models penalized with elastic net (alpha: 0.5; lambda: min; 500 iterations). Elastic net is a type of regularized regression that combines the strengths of Lasso and Ridge regression. It uses variable selection like Lasso and deals with multicollinearity like Ridge. It assigns coefficients only to variables deemed important for explaining the response variable, reducing the risk of overfitting in datasets with more variables than observations, which is the case in this study (and "omic" studies in general) that have a much larger number of variables included than the number of samples (patients) analyzed. Using elastic net as a predictive model-building method does not give confidence intervals or *p*-values, as these are typical of hypothesis-contrast studies, which are not suitable as they are not able to include all the covariates and confounding factors that we are able to include with the current method [16,17].

The models included as covariates for each patient all the SNPs included in the panel, age at diagnosis, type of tumor, drugs received, and the clinical data described in Section 2.2.

## 3. Results

### 3.1. Descriptive Data

#### 3.1.1. Chemotherapy Drugs Administered

Altogether, 24 chemotherapeutic drugs from 37 different protocols (Appendix A) were registered. Considering the 64 genotyped patients, 305 chemotherapy cycles were included (a mean of 4.6 cycles/patient).

#### 3.1.2. Clinical and Analytical Toxicities

Hematologic, digestive, and infectious toxicities were the most frequently registered. The toxicities with a severity degree over or equal to 3 in the CTCAE 4.0 scale and an incidence greater than 5 cases (Figure 2) were: among clinical parameters, fever, neutropenia, and oral mucositis; among analytical parameters, anemia, neutropenia, thrombocytopenia, and hypertransaminasemia (increase of AST, ALT, and GGT). Respiratory toxicity was not frequent (3–7% incidence) but severe in all cases (≥3). Death directly attributable to drug toxicity was not found in the group.

#### 3.1.3. Genotyping

The polymorphisms analyzed in our cohort and the frequency of the variants are shown in Appendix A.

### 3.2. Correlation between Genetic Polymorphisms and Toxicity

Only those clinical variables present in five or more patients with severe toxicity (≥3 on CTCAE 4.0) were included in the correlation study. Thus, the analyzed variables were anemia (43 patients), neutropenia (52 patients), thrombocytopenia (30 patients), hypertransaminasemia (ALT, 15 patients; AST, 15 patients; GGT, 10 patients), neutropenic fever (26 patients), and oral mucositis (7 patients). After elastic net penalized logistic regression, associations were found (Table 1) between severe anemia and 12 polymorphisms (7 with protective effect and 5 with increased risk) of 12 genes and severe neutropenia and 16 polymorphisms (10 with protective effect and 6 with increased risk) of 16 genes. Severe thrombocytopenia was also associated with 18 genetic polymorphisms in 16 genes (9 with a protective effect and 9 with an increased risk).

### 3.3. Correlation between Polymorphisms and Survival

#### 3.3.1. Overall Survival (OS)

Logistic regression evidenced associations between overall survival and 26 genetic polymorphisms belonging to 21 different genes. Among these, 14 are factors of poorer prognosis (increased probability of death during follow-up or lower overall survival: HR higher than 1), whereas the other 12 polymorphisms imply a better prognosis (HR lower than 1) (Table 2).

#### 3.3.2. Event-Free Survival

Associations (Table 2) between event-free survival and 13 polymorphisms in 13 genes were observed. Seven polymorphisms represent a poorer prognosis and increased probability of relapse (HR higher than 1), whereas the other six were associated with a lower risk of disease progression (HR lower than 1).

## 4. Discussion

Whereas most research in the field of cancer chemotherapy efficacy and toxicity has focused on adults, the current paper focuses on an orphan group, infants under 18 months old, using a specially designed pharmacogenetics panel. Particular infant pharmacokinetics and pharmacogenetic characteristics have been considered. The genetic association between SNP variants and therapeutic response, efficacy, and toxicity in oncologic infant patients during a 12-year follow-up at the tertiary Hospital Universitario y Politécnico La Fe (Valencia, Spain) has been evaluated.

Given the nature of the data (62 genetic polymorphisms and 37 types of clinical toxicities vs. 64 patients), which include many more variants than observations, advanced statistical analyses were required, and thus, logistic regression with an elastic net penalty was employed. This statistical method offers robust results. For associations between genetic variants and survival, Cox regression was employed because it considers the temporal evolution.

Associations have been found between 57 variants of 38 SNPs in 32 different genes and the incidence of severe toxicity (anemia, neutropenia, and thrombocytopenia) and survival (overall and event-free). The genes associated encode proteins that transport or metabolize the drugs administered (7 and 17 genes, respectively), have a role in DNA repair and/or control of tumor suppression (5 genes), or other cell functions (4 genes).

The SNPs that showed stronger association with any of the clinical parameters evaluated (toxicities and survival) are marked in bold-type in Table 1 and Table 2 and are summarized based on the function of the proteins encoded in Table 3. These biological functions are drug transporters or metabolizers (7 and 17 genes, respectively); DNA repair and/or control of tumor suppression (5 genes); and other functions (4 genes).

Our results show variants that increase the risk of hematological toxicities and variants that decrease this risk, showing a protective effect. The same happens with prognosis, in terms of overall survival and relapse. For clarity, we will only include in this discussion the findings that increase the risk of toxicity, death, or relapse.

### 4.1. Hematological Toxicities

In our hands, patients carrying rs1801131 GT in *MTHFR* showed the highest increased risk (OR 1.73) of anemia amongst all the SNPs analyzed, which is consistent with other published works, including pediatric oncology scenarios [18]. Regarding neutropenia, we found two SNPs with the highest risk: *C8orf34* rs1517114 GC (OR 1.50) and *XPC* rs2228001 GT (OR 4.63). The second is discussed in the following paragraph, and regarding the first, it has been reported as a predictor of severe diarrhea in irinotecan-treated patients [19], but as far as we know, this is the first work describing increased neutropenia risk. Genetic variants that increase the activity of enzymes responsible for metabolizing drugs that require activation could increase the toxicity of the drug. This occurs with *CYP2B6* and cyclophosphamide [13]. Our study reinforces that rs4802101 TC increases the activity of CYP2B6 and favors the transformation of cyclophosphamide into 4-hydroxy-cyclophosphamide, which is converted to the phosphorous mustard (antineoplastic metabolite) and acrolein, the metabolite that is especially toxic for hematopoietic tissue and urinary apparatus, responsible for very common hemorrhagic cystitis, and in our hands showing an increased risk of thrombocytopenia (OR 1.70). AG at rs1045642 in the *ABCB1* gene has shown a 1.70 OR for thrombocytopenia. This gene encodes glycoprotein-G, a very much studied efflux pump in the field of pharmacogenetics. In the literature, the A allele seems to increase thrombocytopenia risk in the context of pediatric leukemia [20].

*SOD2* encodes superoxide dismutase *2*, a mitochondrial enzyme responsible for diminishing reactive oxygen species and involved in the metabolism of many chemotherapeutic drugs, such as cisplatin, asparaginase, methotrexate, and cyclophosphamide. Variant rs4880 GG has been previously related to ototoxicity [21], but instead we have found an association with thrombocytopenia (OR 1.73). Platelets, in the same manner as cochlear cells, are very vulnerable to free radicals. If the GG variant is causing a decrease in activity of the SOD2 enzyme, the accumulation of ROS could be toxic for the platelets, promoting the observed toxicity. The effects of TNRSF11B are discussed in the following paragraph.

### 4.2. Survival

Establishing a relationship between a polymorphism and prognosis is usually much more complex, since multiple biases and confounding factors interfere. However, we have emphasized those results with stronger associations after employing models that are really stringent. Some of the HR values obtained are quite high, especially considering that we are dealing with a small sample size, so we have to keep in mind that this is an initial pilot study and the results need to be validated in larger populations. Even so, some of our results are supported by previous studies, and of special interest in our group is the relationship between *MTHFR* rs1801133 GG and overall survival. In a previous publication from our group [22], with a totally different cohort of neuroblastoma patients, we already found that this variant was correlated with poorer overall survival in both normal and amplified *MYCN* patients. It is not clear yet if these results are through a direct effect of the enzyme on the chemotherapeutic drugs, or related to DNA methylation and or synthesis, or a combination of all, but recent literature find similar results in different adult and pediatric cancers [18,23,24].

Our findings regarding *TNFRSF11B* rs2073618 GG seem to be new in the context of cancer survival. We found 1.84 HR but also 1.77 OR regarding thrombocytopenia. This gene has been extensively studied in the context of rheumatoid arthritis and osteoporosis, but also with aromatase inhibitors in the context of breast cancer. Being part of the TNF receptor family, we could hypothesize that this signaling pathway could be altered in patients bearing this variant, with potential effects on their immune system tumor surveillance. TNFR2 directly promotes the proliferation of some kinds of tumor cells and, by activating immunosuppressive cells, it supports immune escape and tumor development [25].

Regarding *XPC*, we found decreased overall survival in patients rs2228001 GT (HR 1.68) (also in TT to a lesser extent) and also a remarkably high odds ratio for neutropenia probability (OR 4.63). The literature supports our findings relating the observed effects of platinum compounds on toxicity but also on prognosis in different types of cancer [26,27]. These two effects could in fact be a consequence of each other: if the T allele is really promoting neutropenia, we could expect that the clinician’s response would be a delay in the following chemotherapy cycles and/or drug dose reduction. This "actuation," meaning a decrease in the therapeutic effect, could be the cause of the decreased survival and not the genetic variant itself, but this hypothesis must be confirmed.

*CYP3A4* also showed a variant with a remarkable HR (rs2740574 CT, 2.92). The CYP3A4 enzyme metabolizes doxorubicin, vinblastine, and vincristine, among other drugs, and the T allele in this SNP promotes increased transcription of the gene [28], a higher drug clearance rate, and lower exposure to the drug, which could explain a lower therapeutic effect.

The highest HR found in our study, related to overall survival, belongs to rs4149015 GA in *SLC01B1* impaired transport of many substrates, including methotrexate and irinotecan [29]. This would mean a lower entry into hepatocytes and therefore a reduced clearance of these drugs, which could lead to toxicity and a higher risk of death.

Last but not least, we have two more genes associated with event-free survival. The first is *CDA,* where rs3215400 del.del has been found to be associated with overall survival; it showed 2.19 HR for the first and 2.92 for the later. The encoded enzyme, cytidine deaminase, is directly involved in the inactivation of citarabine, and the selected variant implies a deficient activity. Our patients carrying this variant also showed more neutropenia (OR 1.20). All these effects are in agreement with previously published results both in pediatric and adult populations [30,31].

Our last relevant SNP is located in *SLC19A1*, rs1051266. TT showed a 1.61 HR for event-free survival. This gene encodes RFC-1, or folate transporter protein 1, named for its role in the transport of folate and drugs such as methotrexate (administered to 26.6% of our patients). TT carriers have a worse prognosis, in agreement with the literature, in pediatric leukemia patients and with children and young adults with osteosarcoma, where higher levels of methotrexate are found in the systemic circulation [32].

In our hands, fewer SNPs were found to be associated with EFS compared to those associated with OS. We don’t have a certain explanation for this, but it is true that the patients without any relapse events were 42, while the patients alive were 51 at the end of the follow-up, so we had a larger sample size for the OS analyses, and this could increase the probability of finding associations.

If the associations proposed in this pilot study could be validated in future, larger studies, then the goal would be to use the information in clinical practice. To do so, the inclusion of a high-quality pharmacogenetics report in the electronic medical records seems mandatory as a tool for helping clinicians prescribe in a pharmacogenetics-guided manner. Of course, pediatric oncologists would need to evaluate this new piece of information together with all the rest of the data in order to make a therapeutic decision. In our opinion, only the variants with a higher level of evidence should be included in these clinical pharmacogenetic analyses (those included in drug labels, those included in a relevant clinical implementation guideline, those considered Clinical Annotations Level 1 in PharmGKB, and lastly, those that are not yet in the three previous sources but do have supporting evidence in the literature of a relevant effect on efficacy/toxicity).

## 5. Conclusions

The results of the present work encourage further studies in order to confirm the findings. Increasing scientific contributions support the use of pharmacogenetics as a tool to guide the optimization of treatments, regarding both toxicity and the pharmacological response of patients. Pediatric oncologists are becoming aware that these strategies could make a difference in patients’ clinical management, especially in vulnerable populations. Confirming the findings of this pilot study could bring some of the proposed SNPs to a level of clinical evidence high enough to include them in the guidelines for chemotherapeutic drug use in infants, a population with special vulnerability. The present work is a pilot study, and it has several limitations that we need to keep in mind to prevent unreasonable expectations: It has a small sample size regarding statistics (although a large one if we take into account that the patients included are very rare); it mixes together patients with different tumor types (although the tumor type is included as a covariate to decipher its weight); and the clinical data were retrieved retrospectively from medical records, which is never as accurate as having a prospective design (we tried to guarantee the maximum accuracy with a well-designed clinical data sheet and two different pediatric oncologists collecting the data). We find two steps to confirm these findings: First, a pilot ambispective study very similar to the one presented here, but multicentric at the national or, better yet, international level, to confirm the associations of the SNPs and clinical variants presented here in a larger cohort. The chemotherapy protocols are common in Europe, and this would definitely be an advantage, and we could also start recording long-term follow-up of the patients (cancer survivorship programs, e.g.). After this, with the confirmed SNPs, a more ambitious study could be proposed, acting prospectively, modifying the therapy according to the validated genetic variants in the patients, without altering the drug label specifications of use, and therefore not involving a clinical trial but a real-world quaternary prevention study.

## Figures and Tables

**Figure 1 cancers-15-01424-f001:**
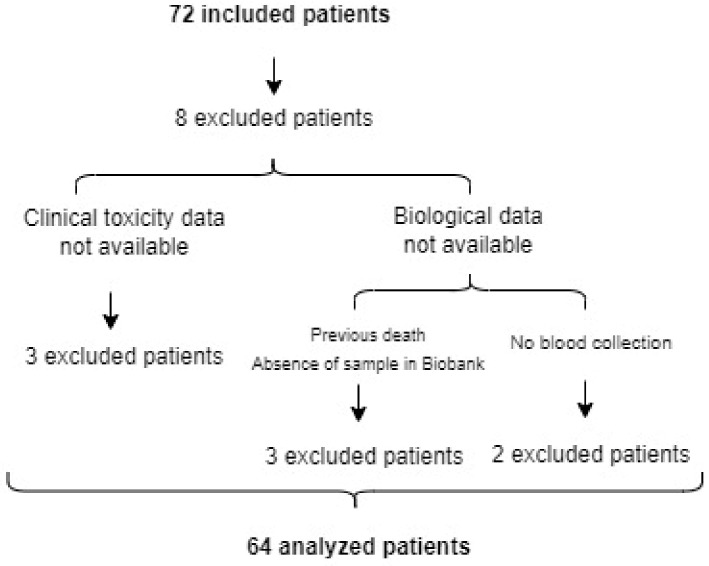
Patients’ inclusion diagram. Patients recruited and those finally included in the study after applying exclusion criteria such as the absence of clinical data or biological data (a blood sample for genotyping).

**Figure 2 cancers-15-01424-f002:**
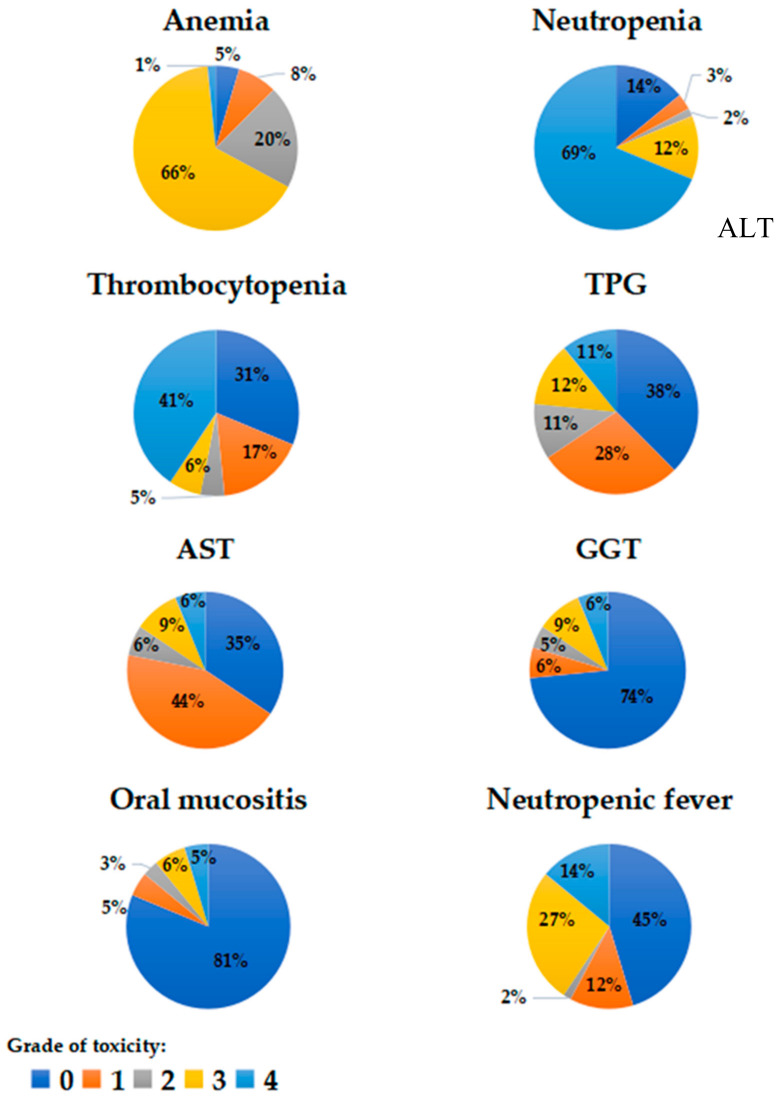
The most common toxicities observed in the study patients during the follow-up period, as well as their severity level according to the CTCAE 4.0 classification. The proportion of patients with anemia, neutropenia, thrombocytopenia, liver enzyme elevations, oral mucositis, and neutropenic fever is shown. The toxicity is graded in five categories according to the CTCAE 4.0 Classification, without any patient achieving grade 5 toxicity. CTAE: Common Terminology Criteria for Adverse Events. ALT: alanine aminotransferase; AST: aspartate aminotransferase; GGT: gamma-glutamyl transferase.

**Table 1 cancers-15-01424-t001:** Polymorphisms associated with toxicities. Those with protective effects are highlighted in green, while those with increased risk are highlighted in red. Panel 1-A includes results of anemia, Panel 1-B neutropenia and Panel 1-C, the results of thrombocytopenia. Bold-type values are those over 1.50 or under 0.66.

1-A ANEMIA			
GENE	SNP	VARIANT	ODDS RATIO
*ATIC*	rs16853826	GG	1.14
** *MTHFR* **	**rs1801131**	**GT**	**1.73**
*DPYD*	rs1801158	TC	1.08
*TNFRSF11B*	rs2073618	GG	1.42
*UGT1A*	rs4124874	TT	1.13
*ABCB1*	rs1045642	GG	0.91
*ITPA*	rs1127354	CC	0.91
*FCGRA1*	rs1801274	AG	0.81
*ESR1*	rs2234693	TT	0.85
*XRCC1*	rs25487	TT	0.87
*CDA*	rs3215400	del.C	0.86
*ATIC*	rs4673993	TT	0.73
**1-B NEUTROPENIA**			
**GENE**	**SNP**	**VARIANT**	**ODDS RATIO**
** *C8ORF34* **	**rs1517114**	**GC**	**1.50**
*MTHFR*	rs1801133	GA	1.02
** *XPC* **	**rs2228001**	**GT**	**4.63**
*CDA*	rs3215400	del.del	1.20
*ABCC2*	rs3740066	TC	1.03
*CYP2C19*	rs4244285	AG	1.15
** *ABCB1* **	**rs1045642**	**GG**	**0.57**
*ERCC1*	rs11615	GG	0.80
*C8ORF34*	rs1517114	GG	0.85
** *ATIC* **	**rs16853826**	**GA**	**0.52**
*FCGR2A*	rs1801274	GG	0.88
*MTRR*	rs1801394	GG	0.75
*XRCC1*	rs25487	TT	0.93
*CDA*	rs3215400	del.C	0.78
*MTR*	rs3768142	TT	0.70
*FCGR3A*	rs396991	CA	0.82
*CYP2B6*	rs4802101	TT	0.85
**1-C THROMBOCYTOPENIA**			
**GENE**	**SNP**	**VARIANT**	**ODDS RATIO**
** *ABCB1* **	**rs1045642**	**AG**	**1.70**
*C8ORF34*	rs1517114	GC	1.45
*MTHFR*	rs1801131	GT	1.41
*MTHFR*	rs1801133	GG	1.32
*MTRR*	rs1801394	AG	1.06
** *TNFRSF11B* **	**rs2073618**	**GG**	**1.77**
*XPC*	rs2228001	GT	1.40
** *CYP2B6* **	**rs4802101**	**TC**	**1.70**
** *SOD2* **	**rs4880**	**GG**	**1.73**
*MTHFR*	rs1801131	TT	0.79
*ATIC*	rs16853826	GA	0.75
*ESR1*	rs2234693	TT	0.77
*CDA*	rs3215400	del.C	0.79
*FCGR3A*	rs396991	CC	0.93
*ATIC*	rs4673993	TT	0.73
*SLC22A1*	rs683369	CG	0.74
** *ABCC2* **	**rs8187710**	**GG**	**0.38**

**Table 2 cancers-15-01424-t002:** Polymorphisms associated with survival variation. The associations of SNP variants with protective effects are shown in green, and those with an increased probability of a lower survival rate are shown in red. Panel 2-A shows the results according to overall survival, and Panel 2-B shows the results according to event-free survival. Bold-type values are those over 1.50 or under 0.66.

2-A GLOBAL SURVIVAL			
*GENE*	SNP	VARIANT	HAZARD RATIO
*ATIC*	rs16853826	GG	1.22
*SLC19A1*	rs1051266	TT	1.06
** *MTHFR* **	**rs1801133**	**GG**	**3.12**
*MTRR*	rs1801394	AG	1.32
** *TNFRSF11B* **	**rs2073618**	**GG**	**1.84**
** *XPC* **	**rs2228001**	**GT**	**1.68**
*XPC*	rs2228001	TT	1.10
*ENOSF1*	rs2612091	CT	1.05
** *CYP3A4* **	**rs2740574**	**CT**	**2.92**
*ERCC1*	rs3212986	CC	1.19
** *CDA* **	**rs3215400**	**del.del**	**1.90**
** *SLC01B1* **	**rs4149015**	**GA**	**3.96**
*CYP2C19*	rs4244285	AG	1.09
*SOD2*	rs4880	AG	1.34
*SLC19A1*	rs1051266	CT	0.70
** *TP53* **	**rs1042522**	**GG**	**0.09**
** *TP53* **	**rs1042522**	**CG**	**0.19**
** *ITPA* **	**rs1127354**	**CC**	**0.59**
** *NQO1* **	**rs1800566**	**GG**	**0.56**
** *MTRR* **	**rs1801394**	**GG**	**0.41**
** *SLC22A2* **	**rs316019**	**CC**	**0.25**
*MTR*	rs3768142	GT	0.95
** *FCGR3A* **	**rs396991**	**CC**	**0.48**
*UGT1A*	rs4124874	GT	0.77
** *SLC01B1* **	**rs4149015**	**GG**	**0.26**
** *FOLH1* **	**rs61886492**	**GG**	**0.15**
**2-B EVENT-FREE SURVIVAL**			
** *GENE* **	**SNP**	**VARIANT**	**HAZARD RATIO**
** *SLC19A1* **	**rs1051266**	**TT**	**1.61**
*UMP*	rs1801019	CG	1.05
*XRCC1*	rs25487	TT	1.39
*ENOSF1*	rs2612091	CT	1.08
*SLC22A2*	rs316019	CC	1.16
** *CDA* **	**rs3215400**	**del.del**	**2.19**
*CYP3A5*	rs776746	CT	1.28
*TP53*	rs1042522	CG	0.85
*SLC01B1*	rs11045879	TT	0.72
*ITPA*	rs1127354	CC	0.67
** *CYP3A4* **	**rs2740574**	**TT**	**0.59**
*SLC01B1*	rs4149056	TT	0.87
** *ABCC2* **	**rs8187710**	**GG**	**0.63**

**Table 3 cancers-15-01424-t003:** Classification of the genes where the more relevant associations between SNPs and toxicity/survival have been found, according to the main function of the protein (transport, metabolism, DNA reparation, tumor suppression, or other biological functions). SLC: Solute carrier family; ABC: ATP-binding cassette family; EFS: event-free survival; OS: overall survival; Tbp: thrombocytopenia; An: anemia; Np: neutropenia.

TRANSPORT PROTEINS			
Solute Carrier Family (SLC)			
**GENE**	**SNP**	**VARIANT**	**EFFECT**
*SLC01B1*	rs4149015	GA-GG	OS
*SLC19A1*	rs1051266	TT	EFS
*SLC22A2*	rs316019	CC	OS
**ATP-binding cassette family (ABC)**			
**GENE**	**SNP**	**VARIANT**	**EFFECT**
*ABCB1*	rs1045642	AG, GG	Tbp, Np
*ABCC2*	rs8187710	GG	Tbp, EFS
**METABOLIC ENZYMES**			
**Phase I**			
**GENE**	**SNP**	**VARIANT**	**EFFECT**
*CYP2B6*	rs4802101	TC	Tbp
*CYP3A4*	rs2740574	TT, CT	EFS, OS
*FOLH1*	rs61886492	GG	OS
*NQO1*	rs1800566	GG	OS
*SOD2*	rs4880	GG	Tbp
*CDA*	rs3215400	del.del	EFS, OS
*ITPA*	rs1127354	CC	OS
**Phase II**			
*MTHFR*	rs1801131	GT	An
	rs1801133	GG	OS
*MTRR*	rs1801394	GG-AG	OS
*ATIC*	rs16853826	GA	Np
**DNA REPAIR GENE/TUMOR SUPPRESSOR GENE**			
**GENE**	**SNP**	**VARIANT**	**EFFECT**
*TP53*	rs1042522	CG-GG	OS
*XPC*	rs2228001	GT	Np, OS
*C8ORF34*	rs1517114	GC	Np
**GENES WITH OTHER FUNCTIONS**			
**GENE**	**SNP**	**VARIANT**	**EFFECT**
*TNFRSF11B*	rs2073618	GG	Tbp, OS
*FCGR3A*	rs396991	CC	OS

## Data Availability

All the data generated in the study is included here.

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
