# Peer review of "Personalized Medicine in Infant Population with Cancer: Pharmacogenetic Pilot Study of Polymorphisms Related to Toxicity and Response to Chemotherapy"

_cancers, 2023, doi:10.3390/cancers15051424_

Round 1

Reviewer 1 Report

This is an interesting paper about the possible effects of pharmacogenomics in young age on hematologic side effects, liver enzymes, oral mucositis, neutropenic fever and survival. The primary idea is excellent, that the metabolism of drugs in young patients could be different compared to older patients, so it could be an important issue. 

This paper also tries to break with the former approach of this problem: altered effectivity of different drugs based on genetic alteration; and different survival in one disease type due to genetic alteration causing altered metabolism of different drugs.

However, to give a general answer which is true for all kind of cancer in young patients is extremely ambitious approach in this study as there are only  64 patients with cca. 10 different diseases and several different chemotherapy approaches. This approach could give statistically significant results, but regarding the heterogeneity of the disease types with highly different survival, heterogeneity of drugs in each patients makes the results highly questionable.

Authors do not share any statistical calculation in terms of survival and histological diagnosis, and side effects and different chemotherapy agents, which highly reduce the credibility of their findings.

Minor comments: 

1. three paragraphs of  Introduction section (line 108-118) are highly irrelevant to the other parts of the paper

Reviewer 2 Report

Thank you for this paper. This is an important exploratory study in an orphan area. Overall, I find the manuscript is well-organized and well-written, with information presented clearly. The figures and tables are well-designed and present information clearly.

Introduction:

This is an issue of writing style, and the authors should feel free to ignore this advice, but I find that the introduction is too long and does not effectively highlight the importance of the study: infants are not mentioned until the sixth paragraph. I suggest rewriting the introduction to make it shorter and to more clearly argue the importance of this study: pharmacogenetics/genomics are important, this is particularly true in pediatric oncology, but this has not been studied in infants.  There is a lot of epidemiological information in the introduction that is not directly related to the present study, and this information can be removed.

Materials and Methods:

2.2.2. For patients in the retrospective cohort, how were toxicity and survival data abstracted? How was the quality of this data abstraction ensured (e.g. independent abstraction of data by a second person)? 

2.4. Elastic net fitting is a concept that may need better explanation for the intended audience for this paper (clinicians, rather than statisticians).  Statements like "...including only variables with statistical predictive power" and "...it does not produce p values. Variables selected in the model are statistically relevant" are somewhat vague. The reader may benefit from some explanation of what the output of this modelling strategy is and how model parameters are determined to be significant. (My understanding is that model coefficients are excluded if they are not statistically likely to different from zero.)

Results:

There appears to be some inconsistency of terminology when discussing GOT, GPT, ALT, AST, etc. Please ensure consistent use of abbreviations.

Discussion:

Page 10, line 283: "...higher levels of significance..." may not be the right phrase to use here, as this sort of terminology can be interpreted to have a particular statistical meaning. The authors may prefer to use "stronger associations" or "larger associations" or similar terminology.

Fewer polymorphisms are associated with EFS than with OS. This observation may be worth discussing.

Page 11, Line 327: I suspect the authors mean "confounding", rather than "confusion".

I find the discussion of possible mechanisms responsible for associations is quite interesting and helpful.

Some discussion of potential limitations of this study is required, particularly with regard to sample size, the primarily retrospective data collection, and the population in which the study was conducted.

It is quite common to make a statement that the findings of a study, particularly a small study or a pilot study, should be confirmed or that further research is required, but it is most helpful when the authors make some comment on what that confirmatory or further study would look like. What sort of studies would the authors suggest be done next?

Similarly, I would be interested to read some comment by the authors on how this information, if confirmed, could be included in clinical practice.

Table S2 appears to have some colour coding of information, but this is not explained in the legend for the table.

Reviewer 3 Report

In their manuscript, Urtasun et al. address an important topic in a very vulnerable population. The topic is of huge importance given the years of life those children have after their treatment. The sample size is amazing given the rareness of the diseases studied in the proposed age group. Statistical analyses have been carried out carefully. The results and discussion section are presented clear and straight forward. Overall, I would like to congratulate the authors for a very interesting article, which is of huge impact and even more enjoyable to read. 

There are only few minor typos or suggestions to change: 

line 52: gene names SLC19A1 and CDA must be in italic. The same is true for Table 3. Please write all gene names in italic. 

line 155: typo: ophthalmological 

line 146 - 147: I cannot find the annotation in the Supplementary Materials. 

line 324: please delete that sentence as it is not necessary for the flow of the manuscript. 

Supplementary Materials, Table S1: 

Typo in the description: OR: Clofarabine; P: Fludarabine; Q: Etoposide; A: Topotecan

Moreover, if you explain the abbreviation Rb, please do so for the others as well.  

Round 2

Reviewer 1 Report

This paper was previously reviewed. Although, authors tried to eliminate the basic problem of this paper by providing more focused Introduction and a wide panel about the methodology and limitation of the study, the basic problems still exist.

I agree with authors that they are in a hard situation, that there is a low number of patients due to rarity of malignant diseases in younger patient. Statistically, results are correct, despite the widely heterogeneity of diseases, heterogeneity of treatments inside each disease type,  different age of patients which is unknown from this paper, but must have a significant effect on side effects and drug metabolism.  

The survival, treatment and side effects are not so simple in this age group as authors try to present this